# Development of Visible/Near-Infrared Hyperspectral Imaging for the Prediction of Total Arsenic Concentration in Soil

**Lifei Wei [1,2], Yangxi Zhang [1,*], Ziran Yuan [1], Zhengxiang Wang [1,2], Feng Yin [3] and Liqin Cao [4]** 

[1]  Faculty of Resources and Environmental Science, Hubei University, Wuhan 430062, China;
    weilifei2508@hubu.edu.cn (L.W.); 201711110811066@stu.hubu.edu.cn (Z.Y.); wangzx66@hubu.edu.cn (Z.W.)
[2]  Hubei Key Laboratory of Regional Development and Environmental Response, Hubei University,
    Wuhan 430062, China
[3]  Hubei Provincial Institute of Land and Resources, Wuhan 430070, China; 201911110811276@stu.hubu.edu.cn
[4]  School of Printing and Packaging, Wuhan University, Wuhan 430079, China; clq@whu.edu.cn
[*]  Correspondence: 201811110811176@stu.hubu.edu.cn

**Abstract:** Soil total arsenic (TAs) contamination caused by human activities—such as mining, smelting, and agriculture—is a problem of global concern. Visible/near-infrared (VNIR), X-ray fluorescence spectroscopy (XRF), and laser-induced breakdown spectroscopy (LIBS) do not need too much sample preparation and utilization of chemicals to evaluate total arsenic (TAs) concentration in soil. VNIR with hyperspectral imaging has the potential to predict TAs concentration in soil. In this study, 59 soil samples were collected from the Daye City mining area of China, and hyperspectral imaging of the soil samples was undertaken using a visible/near-infrared hyperspectral imaging system (wavelength range 470–900 nm). Spectral preprocessing included standard normal variate (SNV) transformation, multivariate scatter correction (MSC), first derivative (FD) preprocessing, and second derivative (SD) preprocessing. Characteristic bands were then identified based on Spearman's rank correlation coefficients. Four regression models were used for the modeling prediction: partial least squares regression (PLSR) ($R^2 = 0.71$, RMSE = 0.48), support vector machine regression (SVMR) ($R^2 = 0.78$, RMSE = 0.42), random forest (RF) ($R^2 = 0.78$, RMSE = 0.42), and extremely randomized trees regression (ETR) ($R^2 = 0.81$, RMSE = 0.38). The prediction results were compared with the results of atomic fluorescence spectrometry methods. In the prediction results of the models, the accuracy of ETR using FD preprocessing was the highest. The results confirmed that hyperspectral imaging combined with Spearman's rank correlation with machine learning models can be used to estimate soil TAs content.

**Keywords:** hyperspectral imaging; soil arsenic; extremely randomized trees regression

## 1. Introduction

Arsenic (As) is a ubiquitous element in nature, and can be found in rocks, soils, sediments, fossil fuels, plants, and almost all living organisms, including the biota of aquatic ecosystems [1]. Worldwide total arsenic (TAs) levels in soils have been reported to range between 2 and 5 mg/kg [2,3]. However, TAs can be very harmful due to excessive accumulation in agricultural soils [4,5]. Firstly, the transfer of TAs from soil to human beings through the food chain poses a potential disease risk [6,7]. Secondly, excess TAs entering the pedosphere can affect the quality of cultivated land and reduce productivity [7,8]. Research has suggested that the TAs can be accumulated due to human activities such as mining and smelting, industrial processes, and agricultural fertilizers. Most countries have been confronted with the soil contamination caused by heavy metals has become a worldwide issue [9–13]. Traditional chemical-based methods are destructive, time-consuming, and expensive. Therefore,

nondestructive, cheap, and rapid methods for detecting soil TAs content, such as hyperspectral imaging, are needed to avoid human health risks and achieve soil protection [14].

Visible/near infrared (VNIR), X-ray fluorescence spectroscopy (XRF), and laser-induced breakdown spectroscopy (LIBS) do not need too much sample preparation and utilization of chemicals to evaluate TAs concentration in soil [15,16]. Hyperspectral imaging utilizes the VNIR spectrum and is used under laboratory conditions to acquire high spectral resolution images of soil, through its advantages of being fast, effective, non-destructive, and low cost [17,18]. Prediction of TAs content is made possible by correlating the spectral data extracted from the hyperspectral images to their corresponding chemical concentrations [19]. Previous studies showed that the partial least squares (PLS) model can be used to determine the TAs concentration in soil samples ($R^2_\text{p}$ = 0.75, $RMSE_\text{p}$ = 153.77) [20].

In recent years, the development of machine learning algorithms also allowed their application for the prediction of element concentration in soil by the use of hyperspectral imaging (400–2500 nm) [21–23]. Compared with support vector machine regression (SVMR), the random forest (RF) model is a more effective machine learning method for developing diagnosis models [23]. Feature selection and machine learning methods are now important methods of predicting total nitrogen, total zinc, and total magnesium [24,25]. Selecting the sensitive bands based on Spearman's rank correlation coefficients is a common approach when estimating soil concentration content [24,25], but there is only a limited number of studies regarding the estimation of TAs content in soil using hyperspectral imaging technology. In addition, distribution maps obtained using hyperspectral imaging techniques are now widely used in agricultural studies, forestry, meat quality testing, etc. [26–29]. Meanwhile, the use of hyperspectral imaging techniques for generating soil TAs concentration distribution maps using machine learning models remains to be studied [30].

The objectives of this study were to investigate the use of VNIR hyperspectral imaging technology in the prediction of TAs concentration in soil. Preprocessing methods were used for selecting the characteristic bands in hyperspectral imaging technology, based on Spearman's rank correlation coefficients. We also compared the machine learning techniques of PLSR, SVMR, RF, and extremely randomized trees regression (ETR) in the prediction of TAs concentration in soil. The estimation of soil TAs content was then achieved based on the best-performing regression model for the prediction of soil TAs concentration distribution map.

## 2. Materials and Methods

### 2.1. Sample Preparation and Soil Chemical Analysis

Soil samples used in this experiment were collected from the area of Daye mine, a typical area of Jianghan Plain in Hubei province, China (114°31′~115°20′E, 29°40′~30°40′N). The climate of this area is subtropical with an annual average temperature of 16.9 °C. According to the classification and codes for Chinese soil (National Standard of China, GB/T 17296-2009), the soils in this area are mainly red soil and yellow-brown soil. The Daye area is a production base of crops and rich in mineral resources [31]. The mining has greatly damaged the ecological environment, and the farmland soil near the mining area has been seriously polluted.

A total of 59 soil samples were collected from different types of cultivated soils near mining areas in Daye. They were taken from the upper soil layer (0–20 cm) in 2018. After removal of the stones and plant roots, then they were sifted through a 200-mesh sieve and then ground into fine particles, approximate particle size after grinding ≤74 μm [32]. Each soil sample was then divided into two parts. One part was sent to the laboratory digested with nitric acid/hydrochloric acid/perchloric acid. After that, measured by atomic fluorescence spectrometry (AFS) (AFS-9730, Haiguang, China) (National Standard of China: analysis of total arsenic contents in soils, GB/T 21191-2007, GB/T 22105.2-2008). Instrument limits of detection (LODs, mg/kg) were 0.001 for TAs. The other part was sent to the dark chamber for hyperspectral imaging (HSI) measurement. The highest observed soil TAs content was

16.41 µg/g and lowest was 7.04 µg/g. The averaged TAs content of soils average was 9.65 µg/g. The soil sample concentrations are listed in Table 1.

**Table 1.** Statistical descriptions for the arsenic content (µg/g) and the soil sample percentages.

| TAs | No. | Maximum | Minimum | Mean | Std. | Skewness | Kurtosis | Per% |
|---|---|---|---|---|---|---|---|---|
| Total data set | 59 | 16.41 | 7.04 | 9.6527 | 1.4699 | 1.74 | 5.71 | 100 |

## 2.2. Hyperspectral Imaging System and Image Acquisition

A VNIR hyperspectral imaging system was used to capture images of the soil samples [33–36] (Figure 1a). The system consisted of the following components: a SNAPSCAN hyperspectral imaging camera (Imec, Belgium), operating in the spectral range of 470–900 nm, with a spectral resolution of 3 nm, producing a total 147 spectral bands; two current-controlled wide spectrum quartz halogen lights; a sample station for scanning; a dark chamber; and data acquisition software (Imec snapscan acquisition, Imec Corp, Leuven, Belgium). Soil samples were positioned on a moving stage and moved into the camera's field of view. Samples were shaken after capturing each image for homogenization, and the imaging was repeated until reproducible spectral signatures were obtained for consecutive images. The acquired imagery (R: 640 nm, G: 548 nm, B: 470 nm) is illustrated in Figure 1b.

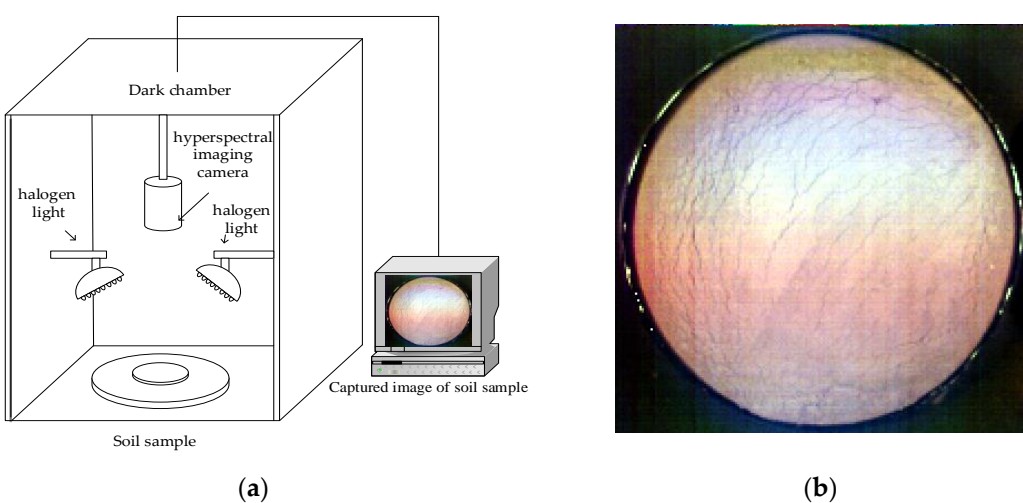

(**a**)  (**b**)

**Figure 1.** (**a**) Hyperspectral imaging setup. (**b**) Acquired imagery (R: 640 nm, G: 548 nm, B: 470 nm).

## 2.3. Spectral Profile Extraction and Data Calibration

To eliminate the impacts of uneven illumination and dark current noise, the raw hyperspectral imagery was calibrated by standard white and dark reference images according to the formula [37]

$$R_c = \frac{R_0 - B}{W - B} \tag{1}$$

where $R_0$ indicates the raw hyperspectral image, $R_c$ represents the calibrated hyperspectral image, $W$ represents the standard white reference image obtained using a rectangular Teflon plate, and $B$ denotes the standard black reference image obtained by covering the lens completely with an opaque black cover [38].

For each hyperspectral image, a region of interest (ROI) was used to measure the mean VNIR spectral reflectance. The ROI (a circle with a diameter of about 150 pixels) was positioned in the middle of the sample image, and close to Petri dish (90 × 17 mm) edge [20,39] (Figure 2). The spectral bands for this study are 519, 560, 564, 576, 697, 700, 703, 706, and 749 nm. The standard deviation of the

averaged spectral band of each sample is between 30~50. The standard deviation of the 697 nm band is even lower than 20.

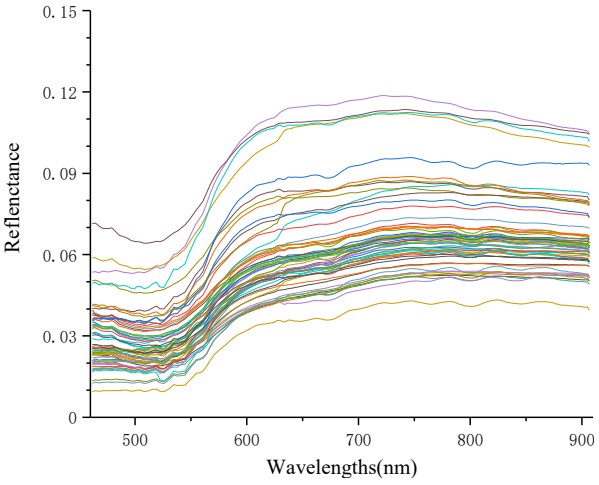

**Figure 2.** Average spectra each of the samples (59 samples).

### 2.4. Feature Band Selection

The reflectance spectral data also contained other irrelevant information and noise. Therefore, before the establishment of the regression model, it was necessary to complete a basic preprocessing to remove the irrelevant information and noise. The common preprocessing methods are first derivative (FD) preprocessing, second derivative (SD) preprocessing, standard normal variate (SNV) transformation, and multivariate scatter correction (MSC) [18,40,41]. We then selected the bands with higher correlation according to the Spearman's rank correlation coefficients [42].

### 2.5. Model Development and Evaluation

In this study, we refer to the detailed of the model information of previous studies, partial least squares regression (PLSR), support vector machine regression (SVMR), random forest (RF), and extremely randomized trees regression (ETR) models were used to analyze the soil sample data. Good results have been obtained in the past based on the PSLR model [20]. SVMR has been proven to be effective in predicting the TAs concentration in soil in many studies [43]. There are many methods for tuning the hyper-parameters of SVMR, grid search being the most frequently used. In this study, use as grid search computes performance at all pairs of e and C to get the performance surface [44]. RF has been reported as performing better than PLSR and SVMR [23,45].

In addition, we also considered the ETR model developed in recent years on the basis of the RF model. The ETR model has been reported as having a higher prediction accuracy than the RF model for soil elements, and has been used in soil spectral prediction models in recent years [46]. ETR was developed as an extension of another tree-based ensemble method (random forest) to be a more computationally efficient algorithm. It consists of three factors: $K$ is the number of randomly selected variables for splitting a node, $n_{min}$ represents the minimum number of samples required for splitting an internal node, and M, the number of trees formed in the ensemble model [47].

According to the results of laboratory measurement samples, the 59 soil samples were divided using the 10-fold cross validation method into calibration set and validation set [48].

The main steps of the work were shown in Figure 3. After hyper spectral image acquisition, correction and reflectance preprocessing, and the ROI spectrum was extractwed. Through the pretreatment methods of FD, SNV and MSC, combined with the Spearman's rank correlation coefficients, the characteristic bands were selected, and the PLSR, SVMR, RF, and ETR models were compared. The best model was used to estimate the soil TAs content and generate the soil TAs distribution map.

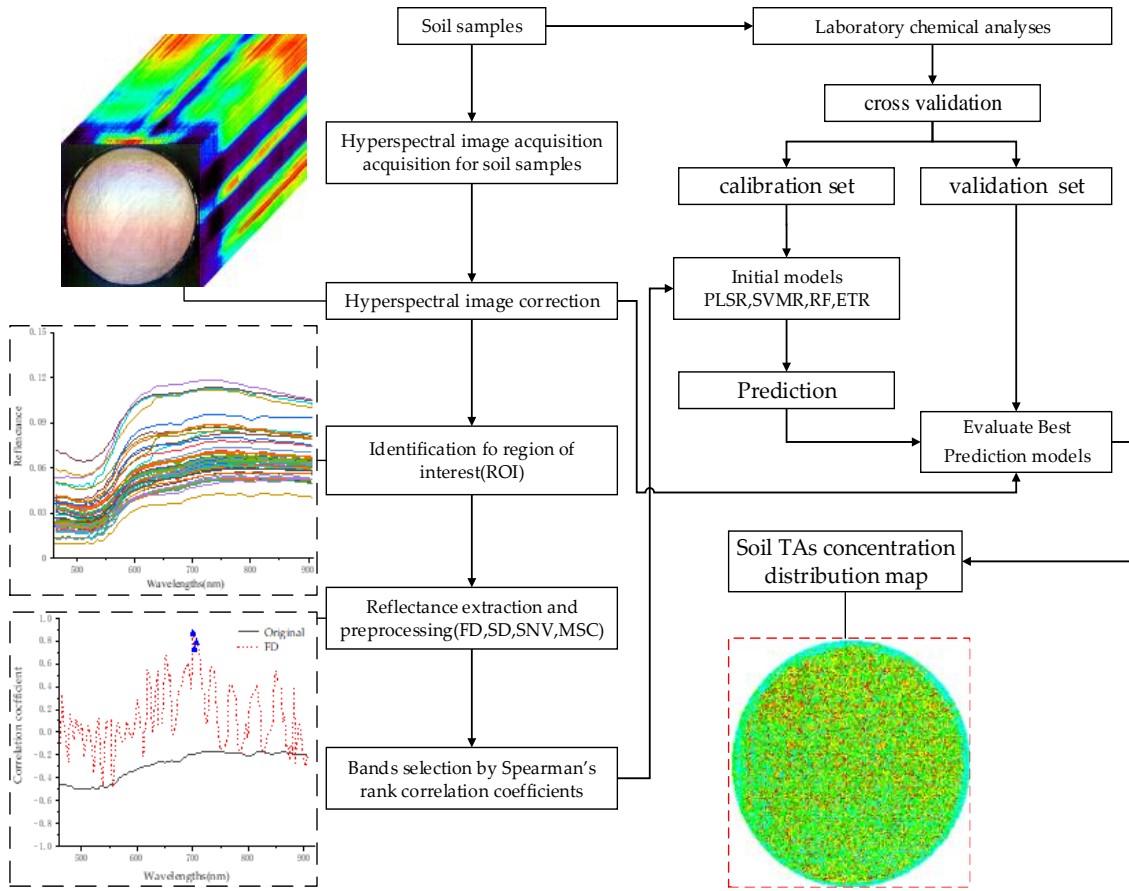

**Figure 3.** Main steps of this work.

The parameters of the determination coefficients ($R^2$), root-mean-square error (RMSE), and relative error (RE%) were used to measure the accuracy of the models [39,49]. The closer $R^2$ is to 1, the better the stability of the model and the higher the degree of fit. RMSE and RE were used to test the predictive ability of the models. The smaller the RMSE and RE, the better the predictive ability.

$$R^2 = 1 - \frac{\sum\limits_{n-1}^{n}(y_i - \hat{y}_i)^2}{\sum\limits_{n-1}^{n}(y_i - \overline{y}_i)^2} \tag{2}$$

$$RMSE = \sqrt{\frac{\sum\limits_{n-1}^{n}(y_i - \hat{y}_i)^2}{n}} \tag{3}$$

$$RE = 100 \times \frac{RMSE}{\overline{y}} \tag{4}$$

where $n$ is the number of samples, $y_i$ is the measured value, $\hat{y}_i$ is the predicted value, and $\overline{y}$ is the average of the measured values.

## 3. Results and Discussion

### 3.1. Preprocessing Comparative Analysis

Feature selection can improve the prediction performance, and gain a better understanding of the data in machine learning. Feature selection by correlation is a commonly used feature selection

method [24,25]. In this study, a Spearman's rank correlation analysis between the TAs concentration in the soil and the preprocessed spectra was carried out.

The preprocessing of the soil spectral can effectively highlight the absorption and reflection bands [18,40,41]. We calculated the Spearman's rank correlation coefficients for the spectral bands after FD, SD, SNV, and MSC preprocessing (Figure 4). In Figure 4, the black line is the original spectrum Spearman's rank correlation coefficients, where it can be seen that the correlation is low, around −0.4 to 0.5. After SNV, MSC, and SD preprocessing, some bands have been improved (red line), but the correlation is still low (0.4 to 0.5). After FD preprocessing, we find bands with correlation of higher than 0.8 appears near the 700 nm region TAs previous studies indicated [20,23,33], the VIS-NIR (650–700 nm) included important wavelengths for estimating TAs contents in soil. Therefore, we selected the three bands (Blue triangle in Figure 4) with the highest correlation coefficient, around 700 nm, and input this into the different machine learning models for calculation. Meanwhile, the three bands (blue triangle in Figure 4) with the highest correlation coefficients in other preprocessing methods are selected for comparison [42].

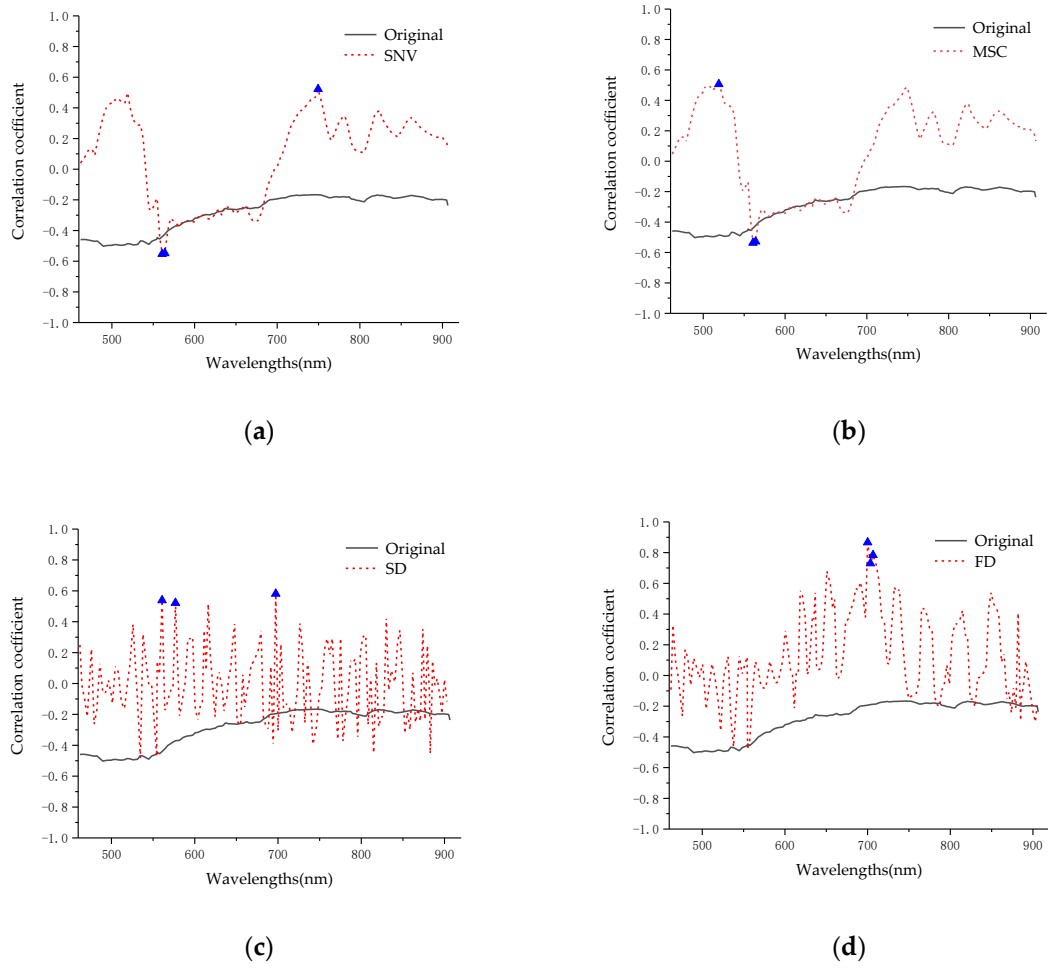

**Figure 4.** Variation of the correlation coefficients after the different preprocessing. (**a**) SNV; (**b**) MSC; (**c**) SD; (**d**) FD.

We then used the model to predict the final result so that we could evaluate the model effect. The results are shown in Table 2. After using SNV, MSC, and SD preprocessing, three bands selected according to the correlation were input into the prediction models, and each model prediction set result was poor. However, the prediction effect of the model with high correlation (0.7 or higher) after using the FD preprocessing is ideal. This shows that as the correlation coefficient increases, the effect of the

model increases. Using a higher correlation band can greatly improve the stability and predictability of the model [50]. Finally, FD was selected as the preprocessing method, and the three characteristic bands with the highest correlation (700, 703, and 706 nm) were selected for the modeling.

**Table 2.** Results of model regression based on the different preprocessing methods.

| Preprocessing and Modeling | Characteristic Bands Wavelength (nm) and Correlation Coefficients | Validation | | |
|---|---|---|---|---|
| | | $R^2_{CV}$ | $RMSE_{CV}$ | $RE_{cv}$ (%) |
| SNV+PLSR | | 0.49 | 0.63 | 6.63 |
| SNV+SVMR | 560.7 (−0.55), 564.0 (−0.54), 749.5 (0.52) | 0.56 | 0.58 | 6.14 |
| SNV+RF | | 0.53 | 0.60 | 6.31 |
| SNV+ETR | | 0.59 | 0.56 | 5.86 |
| MSC+PLSR | | 0.07 | 0.76 | 7.88 |
| MSC +SVMR | 519 (0.50), 560.7 (−0.53), 564.0 (−0.52) | 0.23 | 0.69 | 7.22 |
| MSC +RF | | 0.25 | 0.68 | 7.15 |
| MSC +ETR | | 0.26 | 0.67 | 6.98 |
| SD+PLSR | | 0.23 | 0.64 | 6.64 |
| SD +SVMR | 560.7 (0.53), 576.7 (0.52), 697.0 (0.58) | 0.36 | 0.58 | 6.09 |
| SD +RF | | 0.51 | 0.51 | 5.47 |
| SD +ETR | | 0.51 | 0.51 | 5.43 |
| FD+PLSR | | 0.71 | 0.48 | 5.03 |
| FD +SVMR | 700 (0.86), 703 (0.72), 706 (0.78) | 0.78 | 0.42 | 4.50 |
| FD +RF | | 0.78 | 0.42 | 4.45 |
| FD +ETR | | 0.81 | 0.38 | 4.08 |

### 3.2. Regression Model

PLSR, SVMR, RF, and ETR were used to model the regression. The calibration set was used to train the prediction of the TAs concentration in the soil model. Comparing the model predictions with the validation sets, it can be seen from Figure 5 that the four models all obtain a good accuracy. Model predictive power is estimated by the $R^2$. The closer the value of $R^2$ to 1, and closer the scatter plot of the measured value and predicted accuracy value for the 1:1 line. Among them, the ETR regression model shows the smallest deviation from the 1:1 line, and the degree of fitting is the highest. Model predictive accuracy estimated by the RMSE and RE (%). Most of the predictions are closely distributed around the 1:1 line, few predictions far away from the 1:1 line generated errors, indicating that the models are accurate. From the results of the model accuracy evaluation, the RMSE of the ETR (RMSE = 0.38) model is the lowest, RE (%) of the ETR (RE = 4.08%) model is also lower, indicating that the ETR prediction accuracy is optimal.

From Table 3, it can be seen that the PLSR prediction is lowest, and the $R^2$, RMSE, and RE (%) of the validation set are 0.71, 0.48, and 5.03, respectively. Meanwhile, the $R^2$, RMSE, and RE (%) of the ETR validation set are 0.81, 0.38, and 4.08, respectively, which represent the best prediction results for prediction models. In summary, ETR has advantages in four models of model prediction power and prediction accuracy.

**Table 3.** Accuracy validation of the different models.

| Modeling Method | $R^2_{cv}$ | $RMSE_{cv}$ | $RE_{cv}$ (%) |
|---|---|---|---|
| PLSR | 0.71 | 0.48 | 5.03 |
| SVMR | 0.78 | 0.42 | 4.50 |
| RF | 0.78 | 0.42 | 4.45 |
| ETR | 0.81 | 0.38 | 4.08 |

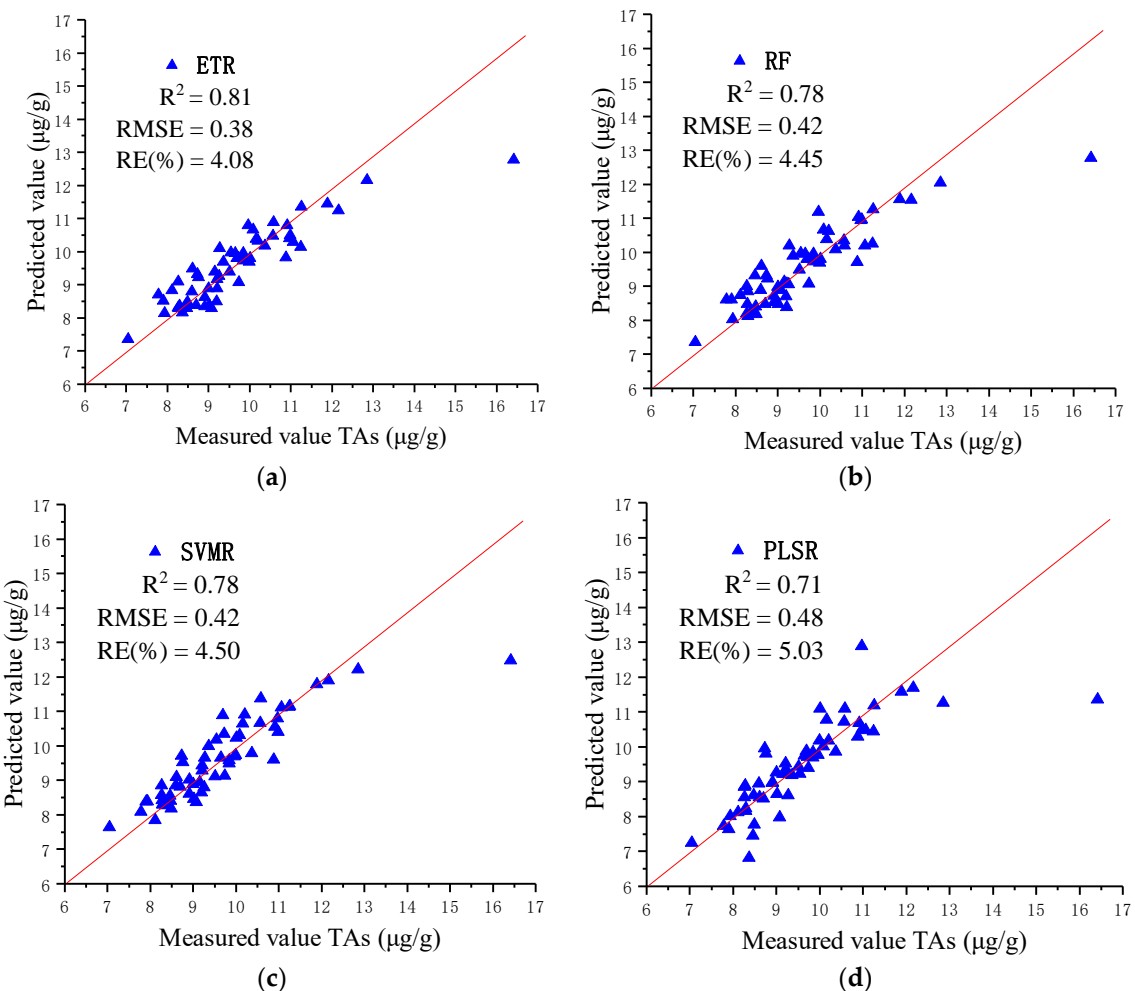

**Figure 5.** Comparison between the measured values and predicted values of the different regression models. (**a**) ETR; (**b**) RF; (**c**) SVMR; (**d**) PLSR.

*3.3. Concentration Distribution Map*

The superiority of hyperspectral images to simultaneously obtain both spectral and spatial information makes it possible to display the results of soil TAs concentration distribution map. This study picks eight soil sample hyperspectral images to generate a TAs concentration distribution map, the maximum TAs concentration soil sample (Figure 6h), minimum TAs concentration soil sample (Figure 6a), and 6 other TAs soil samples. The best model—ETR based on FD preprocessed—was selected to visualize the soil TAs concentration distribution map. The spectral information on each pixel in hyperspectral images was input into ETR model to predict the results. Combined with the prediction results of spatial location information of hyperspectral images, the TAs concentration distribution map could be eventually formed [37,51]. Then, according to the prediction values, divide the values in the graph into five intervals to statistical analysis, cyan (0–8 µg/g), green (8–10 µg/g), yellow (10–12 µg/g), orange (12–14 µg/g), and red (14+ µg/g) (Figure 6).

In Figure 6, with more cyan and green sample distribution maps, the sample prediction value is low. Sample distribution maps with more red, orange, and yellow have higher predicted values. According to the results in the Figure 6, mean and standard deviation were statistically analyzed, as shown in Table 4.

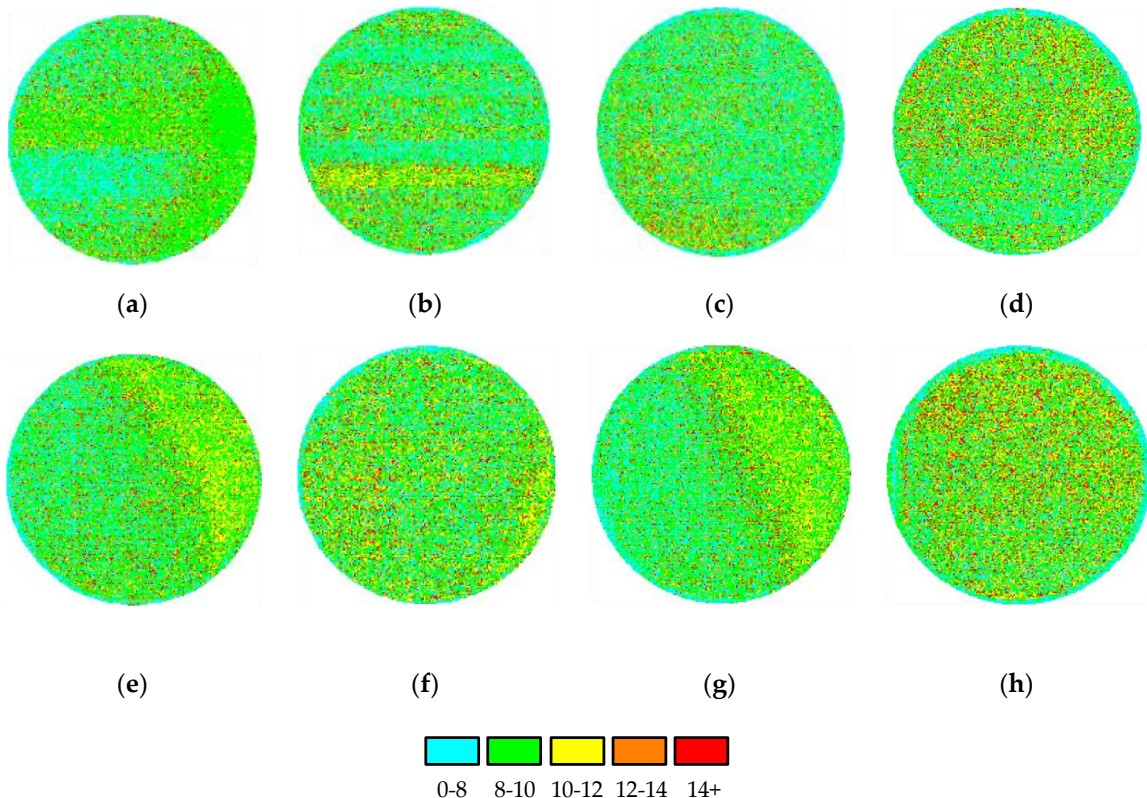

**Figure 6.** Soil TAs concentration distribution maps (Label value is soil sample of arsenic concentration measured with atomic fluorescence spectrometry). (**a**) Sample a 7.04 (µg/g); (**b**) sample b 8.26 (µg/g); (**c**) sample c 8.69 (µg/g); (**d**) sample d 9.36 (µg/g); (**e**) sample e 10.58 (µg/g); (**f**) sample f 11.05 (µg/g); (**g**) sample g 11.25 (µg/g); (**h**) sample h 16.41 (µg/g).

**Table 4.** Statistical summary of the TAs distribution maps.

| No. | Measured Value (µg/g) | Std. | Mean | 0–8 (µg/g) | 8–10 (µg/g) | 10–12 (µg/g) | 12–14 (µg/g) | 14+ (µg/g) |
|-----|------------------------|------|------|-----------|-------------|--------------|--------------|------------|
| a | 7.04 | 4.10 | 8.01 | 37% | 42% | 4% | 13% | 4% |
| b | 8.26 | 4.12 | 8.58 | 32% | 47% | 5% | 11% | 5% |
| c | 8.69 | 4.13 | 8.59 | 25% | 56% | 7% | 8% | 4% |
| d | 9.36 | 4.20 | 8.63 | 23% | 54% | 5% | 11% | 7% |
| e | 10.58 | 4.23 | 8.68 | 24% | 50% | 10% | 9% | 6% |
| f | 11.05 | 4.36 | 8.92 | 23% | 51% | 9% | 11% | 6% |
| g | 11.25 | 4.37 | 8.96 | 25% | 48% | 12% | 10% | 5% |
| h | 16.41 | 4.39 | 9.05 | 22% | 48% | 9% | 13% | 8% |

Shown in Table 4, all soil sample maps with low values (0–10) µg/g (cyan and green) totaled around 75%. The higher area (more than 10 µg/g) (yellow, orange, red) is about 25%. Meanwhile, according to Table 4, HSI predicted TAs and measured TAs used the standard deviation plotted as error bars for each sample is drawn (Figure 7).

As shown in Figure 7, the mean HSI predicted value increases as the measured value increases, confirming a positive correlation between the two datasets. Furthermore, on the whole, the HSI and measured values are in agreement when considering the standard deviation associated with the HSI prediction.

Overall, the results of TAs content in soil samples for measured value were compared with the results shown in the distribution map. The results show that the concentration gradually increases

from soil sample a to soil sample h. This confirms that the model is highly correlated with the real results. It shows that the soil TAs content distribution map generated by the model is valid.

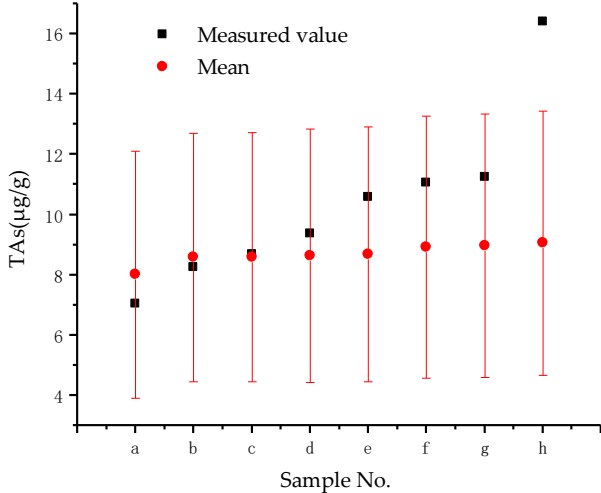

**Figure 7.** HSI predicted TAs and measured TAs with the standard deviation as error bars.

## 4. Conclusions

In this study, we collected 59 soil samples from the Daye City mining area of China. Hyperspectral imaging of the soil samples was undertaken using a hyperspectral imaging system (470–900 nm). Through the pretreatment methods of FD, SNV, and MSC, combined with the Spearman's rank correlation coefficients, the characteristic bands were selected, and the PLSR, SVMR, RF, and ETR models were compared. The ETR model was used to estimate the soil TAs content and generate the soil TAs distribution map. The main conclusions are as follows:

(1) Using the images acquired in the hyperspectral imaging system, bands selected according to different correlation coefficients are put into different models for prediction, it was found that the Spearman's rank correlation coefficients were an effective way to select the characteristic bands of TAs content. ETR ($R^2 = 0.81$, RMSE = 0.38), RF ($R^2 = 0.78$, RMSE = 0.42), SVMR ($R^2 = 0.78$, RMSE = 0.42) models are capable of predicting total As content.

(2) Soil TAs concentration distribution map shows, the Spearman's rank correlation coefficients selected bands for ETR model, to predict the soil TAs distribution map generated by the pixel spectral of the hyperspectral image can be used as for estimation of TAs concentration in soil.

The restriction on estimating total As could be considered a limitation of this present study. This is because not all forms of As are soluble and thus toxic. Therefore, in the context of toxicity, future research should focus instead on predicting the concentration of bioavailable As.

**Author Contributions:** L.W. and Y.Z. were responsible for the overall design of the study and contributed to the proofreading of the manuscript. Z.Y. performed the experiments. Y.Z. analyzed and interpreted the data and wrote the manuscript. L.C. and F.Y. helped with the proofreading of the manuscript. Z.W. contributed to designing the study and the proofreading of the manuscript. All authors read and approved the final manuscript.

**Funding:** This research was funded by the "National Key Research and Development Program of China" (2019YFB2102902, 2017YFB0504202), the "Open Fund of Key Laboratory of Urban Land Resources Monitoring and Simulation, MNR" (KF-2019-04-006), the Opening Foundation of State Key Laboratory of Geo-Information Engineering (SKLGIE2018-M-3-3),the Central Government Guides Local Science and Technology Development Projects (2019ZYYD050), the Opening Foundation of Hunan Engineering and Research Center of Natural Resource Investigation and Monitoring(2020-2), the "Open Fund of the State Laboratory of Information Engineering in Surveying, Mapping, and Remote Sensing, Wuhan University" (18R02) and the "Open fund of Key Laboratory of Agricultural Remote Sensing of the Ministry of Agriculture" (20170007).



**Acknowledgments:** We gratefully acknowledge the help of the Data Extraction and Remote Sensing Analysis Group of Wuhan University (RSIDEA) in collecting the data. The Remote Sensing Monitoring and Evaluation of Ecological Intelligence Group of Hubei University (RSMEEI) helped to process the data. In addition, we are grateful to Mark Ackerley for the English editing.

**Conflicts of Interest:** The authors declare no conflicts of interest.

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
