# Peer review of "Development of Visible/Near-Infrared Hyperspectral Imaging for the Prediction of Total Arsenic Concentration in Soil"

_applsci, doi:10.3390/app10082941_

Round 1

Reviewer 1 Report

This paper has made substantial improvements and a lot of the questions that I raised in my previous review have been addressed. However, the crucial question #4 (i.e. about fig 6) in the previous review has not been addressed properly:

  1. The new pie chart presentation to represent the distribution of the As content in the soil over 5 ranges of concentration is a good idea, however, it is still not obvious to the reader how this can be visualised from fig 6. So it is suggested to perform the following:
    1. Remove a:d, and i:L and present each of them into 5 different colours false colour map to represent the 5 bands of concentrations as that in the pie chart (ie, 0-8, 8-10, 10-12, 12-14, 14+). This can be done easily using say Matlab. This will give the reader at least visual understanding of how the As content predicted by HSI vs the ground truth (GT) measurement.
    2. To compile some statistics, eg, mean or weighted mean, standard deviation (STD) etc of the HSI prediction (ie fig 6 a:d & i:L) for each sample, then plot a graph of HSI prediction vs GT with error bars (eg STD) to see if there is a linear relationship.
    3. Instead of using the pie chart which is difficult to visualise over the 8 samples, it may be better to re-present them into a table form.

Author Response

Response to Reviewer 1 Comments

Thank you for your valuable comments, your comments will guide us in the right direction for our next research, the comments have been carefully took into account and a new revised submission have been uploaded. We highlighted all the altered passages, the author made the following answers and revisions based on the review comments.

The new pie chart presentation to represent the distribution of the As content in the soil over 5 ranges of concentration is a good idea, however, it is still not obvious to the reader how this can be visualised from fig 6. So it is suggested to perform the following:

Remove a:d, and i:L and present each of them into 5 different colours false colour map to represent the 5 bands of concentrations as that in the pie chart (ie, 0-8, 8-10, 10-12, 12-14, 14+). This can be done easily using say Matlab. This will give the reader at least visual understanding of how the As content predicted by HSI vs the ground truth (GT) measurement.

Response 1: thank you for thils valuabe comments, As your opinion. We removed the pie chartsThen HSI re-created the false color map according to 5 different intervals(0-8, 8-10, 10-12, 12-14, 14+).And rewrite the entire paragraph.

To compile some statistics, eg, mean or weighted mean, standard deviation (STD) etc of the HSI prediction (ie fig 6 a:d & i:L) for each sample, then plot a graph of HSI prediction vs GT with error bars (eg STD) to see if there is a linear relationship.

Response 2: thank you for thils valuabe comments, As your opinion. We calculated the mean and standard deviation of the HSI data, and plotted a linear relationship with the ground truth. And rewrite the entire paragraph.

Instead of using the pie chart which is difficult to visualise over the 8 samples, it may be better to re-present them into a table form.

Response 3: thank you for thils valuabe comments, As your opinion. We removed the pie charts and used tables instead of statistical data. And rewrite the entire paragraph

Reviewer 2 Report

The comments have been addressed. Please accept in the present form

Author Response

Thank you for your valuable comments。Thank you again for your review work.

Reviewer 3 Report

Comments to “Development of Visible–Near Infrared Hyperspectral Imaging for the Prediction of Arsenic Concentration in Soil”

The submitted manuscript reports results regarding the utilization of Vis-NIR Hyperspectral Imaging to estimate TOTAL arsenic (As) concentration in soil samples.

Although the topic is not completely new, in my opinion it is still of great interest for Applied Sciences readership. The paper is generally well organized, balanced and to the point, although not often well written (the paper needs to be revised by an English native speaker).

GENERAL COMMENTS

The authors should clearly report throughout the paper that what they are determining is total As.

The authors should clearly replace throughout the paper “ug/g” with “µg/g”.

ABSTRACT

The abstract should be improved. In particular:

The authors state that “The common methods of determining arsenic (As) concentration in soil is expensive and time-consuming” (lines 17-18). This is not true…e.g., XRF. I think that what you need to underline that techniques like Vis-NIR, XRF, LIBS do not need too much sample preparation and utilization of chemicals…

The authors state that “The prediction results were compared with the results of conventional laboratory measurement methods” (lines 30-31). This is not true. Conventional lab techniques to determine As concentration in soil are ICP-OES, ICP-MS, XRF, AAS, etc. Therefore, the authors should simply report here the technique they used.

INTRODUCTION

The Introduction misses some very important general info, including total As concentration in soils worldwide, main sources, forms, etc etc (see suggested references). For example, they should at least report that As concentration in the upper crust ranges between 2 and 5 mg/kg (Wedepohl, 1995; Rudnick and Gao, 2005), or that the median total As content in soil is around 6 mg/kg (Chester, 2000).

While they mentioned in the abstract that there are other, common methods to determine As (that “are expensive and time-consuming”; lines 17-18), they did not stress this concept in the Introduction… For example, the authors could underline that techniques like Vis-NIR, XRF, LIBS do not need too much sample preparation and utilization of chemicals (e.g., Cheburkin and Shotyk, 1996; Dell’Aglio et al., 2011). At the same time, the authors should also mention some possible limitation of the technique.

MATERIALS & METHODS

The authors digested samples using HNO3/HCl/HClO4 (line 93). Is there a reference for this digestion? Is it assumed to mirror total As?

At line 94, the author state that the solution “was then measured with potassium borohydride/silver nitrate spectrophotometry”. Although I work with metals, to be honest, I never heard about this technique… Could you please provide more details (e.g., LOD, LOQ, standards, CRM, equipment, etc)? Why the authors did not use more common and well-known techniques including ICP-OES, ICP-MS, AAS or XRF?

RESULTS AND DISCUSSION

The authors should at least mention the limitations of this study. In particular:

  1. A) The authors should clearly specify that what they are determining is total As and not the available one. In most of the cases, for eco-toxicological studies, researchers look for bioavailable As, not for total… I mean, there are As forms (e.g., FeAsS) that are not soluble (or have a very low solubility) and thus not toxic.
  2. B) The authors tested soil samples having a range of As concentration between 7 and 16 mg/kg. As I reported above, the median As concentration in topsoil is around 6 mg/kg, with lower concentration values reaching 0.3 mg/kg. Do you have an idea about the LOQ of your technique?
  3. C) You tested your approach in As contained soil samples… Do you think your approach will work when, besides As, soil samples show also high concentrations of other elements (e.g., Pb)? What about possible interferences?

REFERENCES

The journal name is missing for Ref. 9, 20, 21, 22, 23, 27, 39, 42

DETAILED (MINOR) COMMENTS.

line 16: “…mining, smelting and agriculture, is…”

line 34: replace “estimation” with “estimate”

line 40: I would suggest to replace ref. [1] (an abstract) with more relevant works (e.g., Kabata-Pendias and Pendias, 2001; Miano et al., 2014)

lines 42-45: I would suggest to add some more references, e.g., Kabata-Pendias and Mukherjee, 2007; Miano et al., 2014

line 59: “element”? Please report which elements…i.e., C and N, not metals

line 89: replace “soil zone” with “soils”

line 90: replace “from 2018” with “in 2018”

line 92: why “<74 µm”?

line 94: replace “It” with “The solution”

line 125: replace “A” with “a”

line 238: “…in soil…” (x2)

line 247: why from 0 if the min. concentration is 7 µg/g?

Consequently, the manuscript cannot be accepted in the present form, but needs moderate revisions.

Suggested references:

Cheburkin AK, Shotyk W (1996) An Energy-dispersive Miniprobe Multielement Analyzer (EMMA) for direct analysis of Pb and other trace elements in peats. Fresenius' Journal of Analytical Chemistry 354, 688–691.

Chester R (2000) Marine Geochemistry, 2nd Ed. Blackwell Science Ltd.

Dell’Aglio M, Gaudiuso R, Senesi GS, De Giacomo A, Zaccone C, Miano TM, De Pascale O (2011) Monitoring of Cr, Cu, Pb, V and Zn in polluted soils by laser induced breakdown spectroscopy (LIBS). Journal of Environmental Monitoring, 13: 1422-1426.

Kabata-Pendias A, Mukherjee AB (2007) Trace elements from soil to humans. Springer, Berlin.

Kabata-Pendias A, Pendias H (2001) Trace elements in soils and plants, 3rd Ed. CRC Press, Boca Raton.

Miano T, D’Orazio V, Zaccone C (2014) Chapter 9. Trace elements and food safety. In: PHEs, Environment and Human Health. Potentially harmful elements in the environment and the impact on human health, pp. 339-370, Springer (doi: 10.1007/978-94-017-8965-3_9).

Rudnick RL, Gao S (2005) Composition of the continental crust. In: The Crust, Volume 3: Treatise on Geochemistry. Elsevier-Pergamon, Amsterdam, 1-64.

Wedepohl KH (1995) The composition of the continental crust. Geochimica Cosmochimica Acta 59, 1217−1232.

Author Response

Response to Reviewer 3 Comments

Thank you for your valuable comments, your comments will guide us in the right direction for our next research, the comments have been carefully took into account and a new revised submission have been uploaded. We highlighted all the altered passages, the author made the following answers and revisions based on the review comments.

The authors should clearly report throughout the paper that what they are determining is total As.

Response : thank you for thils valuabe comments, As your opinion.The title has been revised to “total arsenic”. meanwhile, In this article replace all the paper “AS” with (total arsenic)TAs.

The authors should clearly replace throughout the paper “ug/g” with “µg/g”.’

Response : thank you for thils valuabe comments, As your opinion.All “ug / g” in the text have been replaced with “µg / g”. ’

The abstract should be improved. In particular:

The authors state that “The common methods of determining arsenic (As) concentration in soil is expensive and time-consuming” (lines 17-18). This is not true…e.g., XRF. I think that what you need to underline that techniques like Vis-NIR, XRF, LIBS do not need too much sample preparation and utilization of chemicals…

Response : thank you for thils valuabe comments, As your opinion. We revised this part as

“Soil arsenic (As) contamination caused by human activities, such as mining, smelting and agriculture, is a problem of global concern. (Visible–Near Infrared)Vis-NIR, XRF, LIBS do not need too much sample preparation and utilization of chemicals evaluate total arsenic(TAs)concentration in soil. Vis-NIR with hyperspectral imaging has the potential to predict the TAs concentration in soil.”

authors state that “The prediction results were compared with the results of conventional laboratory measurement methods” (lines 30-31). This is not true. Conventional lab techniques to determine As concentration in soil are ICP-OES, ICP-MS, XRF, AAS, etc. Therefore, the authors should simply report here the technique they used.

Response : thank you for thils valuabe comments, As your opinion. We revised this part as

 “The prediction results were compared with the results of atomic fluorescence spectrometry methods.”

The Introduction misses some very important general info, including total As concentration in soils worldwide, main sources, forms, etc etc (see suggested references). For example, they should at least report that As concentration in the upper crust ranges between 2 and 5 mg/kg (Wedepohl, 1995; Rudnick and Gao, 2005), or that the median total As content in soil is around 6 mg/kg (Chester, 2000).

Response : thank you for thils valuabe comments, As your opinion. We revised this part as add

“Worldwide total As levels in soils have been reported to range between 2 and 5 mg/kg”

And add ref.

Wedepohl, K. The Composition of the Continental Crust. Geochim. Cosmochim. Acta 1995, 59, 1217-1232, doi:10.1016/0016-7037(95)00038-2.

Rudnick, R.; Gao, S. Composition of the Continental Crust. Treatise Geochem 3:1-64. Treatise on Geochemistry 2003, 3, 1-64, doi:10.1016/B0-08-043751-6/03016-4.

While they mentioned in the abstract that there are other, common methods to determine As (that “are expensive and time-consuming”; lines 17-18), they did not stress this concept in the Introduction… For example, the authors could underline that techniques like Vis-NIR, XRF, LIBS do not need too much sample preparation and utilization of chemicals (e.g., Cheburkin and Shotyk, 1996; Dell’Aglio et al., 2011). At the same time, the authors should also mention some possible limitation of the technique.

Response : thank you for thils valuabe comments, As your opinion. We revised this part as add

 “Vis-NIR, XRF, LIBS do not need too much sample preparation and utilization of chemicals evaluate TAs concentration in soil”

ref

Cheburkin, A.; Shotyk, W. An Energy-dispersive Miniprobe Multielement Analyzer (EMMA) for direct analysis of Pb and other trace elements in peats. Anal. Bioanal. Chem. 1996, 354, 688-691, doi:10.1007/s0021663540688.

  1. Dell'Aglio, M.; Gaudiuso, R.; Senesi, G.S.; De Giacomo, A.; Zaccone, C.; Miano, T.M.; De Pascale, O. Monitoring of Cr, Cu, Pb, V and Zn in polluted soils by laser induced breakdown spectroscopy (LIBS). Journal of Environmental Monitoring 2011, 13, 1422-1426.

The authors digested samples using HNO3/HCl/HClO4 (line 93). Is there a reference for this digestion? Is it assumed to mirror total As?

At line 94, the author state that the solution “was then measured with potassium borohydride/silver nitrate spectrophotometry”. Although I work with metals, to be honest, I never heard about this technique… Could you please provide more details (e.g., LOD, LOQ, standards, CRM, equipment, etc)? Why the authors did not use more common and well-known techniques including ICP-OES, ICP-MS, AAS or XRF?

Response : thank you for thils valuabe comments. We are very sorry that did not describe clearly. We have add the name of the experiment method.

Then revised this part

“One parts was sent to the laboratory measured by atomic fluorescence spectrometry (AFS) (National Standard of China, GB/T 21191-2007).”

The authors should at least mention the limitations of this study. In particular:

  1. A) The authors should clearly specify that what they are determining is total As and not the available one. In most of the cases, for eco-toxicological studies, researchers look for bioavailable As, not for total… I mean, there are As forms (e.g., FeAsS) that are not soluble (or have a very low solubility) and thus not toxic.

Response : thank you for thils valuabe comments. Based on your suggestions, change the arsenic in the conclusion to “total arsenic”

  1. B) The authors tested soil samples having a range of As concentration between 7 and 16 mg/kg. As I reported above, the median As concentration in topsoil is around 6 mg/kg, with lower concentration values reaching 0.3 mg/kg. Do you have an idea about the LOQ of your technique?

Response : thank you for thils valuabe comments. The main study objective is to assess soil pollution. The study area we selected is a suspected pollution area, so it will be higher than the average. Because the machine learning algorithm we use has a correlation with the ground truth, it also worked in predicting lower values.

  1. C) You tested your approach in As contained soil samples… Do you think your approach will work when, besides As, soil samples show also high concentrations of other elements (e.g., Pb)? What about possible interferences?

Response : thank you for thils valuabe comments.In the article, I have used the method of spectrum preprocessing to process the spectrum.eg FD, SD, etc.,these can eliminate some interferences.in Section 3.1

REFERENCES

The journal name is missing for Ref. 9, 20, 21, 22, 23, 27, 39, 42

Response : As your opinion.we revise  the missing information of Ref. 9, 20, 21, 22, 23, 27, 39, 42

DETAILED (MINOR) COMMENTS.

line 16: “…mining, smelting and agriculture, is…”

Response : As your opinion.we revise to” Soil total arsenic (TAs) contamination caused by human activities, such as mining, smelting and agriculture, is a problem of global concern.”

line 34: replace “estimation” with “estimate”

Response : As your opinion.we revise  to “estimate”

line 40: I would suggest to replace ref. [1] (an abstract) with more relevant works (e.g., Kabata-Pendias and Pendias, 2001; Miano et al., 2014)

Response : As your opinion.we replace and add ref to

 Kabata-Pendias A, Pendias H (2001) Trace elements in soils and plants, 3rd Ed. CRC Press, Boca Raton.

Miano T, D’Orazio V, Zaccone C (2014) Chapter 9. Trace elements and food safety. In: PHEs, Environment and Human Health. Potentially harmful elements in the environment and the impact on human health, pp. 339-370, Springer (doi: 10.1007/978-94-017-8965-3_9).

lines 42-45: I would suggest to add some more references, e.g., Kabata-Pendias and Mukherjee, 2007; Miano et al., 2014

Response : As your opinion.we add ref

Kabata-Pendias A, Mukherjee AB (2007) Trace elements from soil to humans. Springer, Berlin.

Miano T, D’Orazio V, Zaccone C (2014) Chapter 9. Trace elements and food safety. In: PHEs, Environment and Human Health. Potentially harmful elements in the environment and the impact on human health, pp. 339-370, Springer (doi: 10.1007/978-94-017-8965-3_9).

line 59: “element”? Please report which elements…i.e., C and N, not metals

Response : As your opinion.we  revise  “enture selection and machine learning methods are now important methods of predicting total nitrogen, zinc magnesium[24,25]”

line 89: replace “soil zone” with “soils”

Response : As your opinion.we  revise  to “soils”

line 90: replace “from 2018” with “in 2018”

Response : As your opinion.we  revise  to “in 2018”

line 92: why “<74 µm”?

Response : As your opinion.we  revise  “um” to “µm”.the size information is added according to the requirements of other reviewers.

line 94: replace “It” with “The solution”

Response : As your opinion.we  revise  to “The solution”

line 125: replace “A” with “a”

Response : As your opinion.we  revise  to “a”

line 238: “…in soil…” (x2)

Response : The relevant paragraphs have been deleted and rewritten due to requests from other reviewers. 

line 247: why from 0 if the min. concentration is 7 µg/g?

 Response : The relevant paragraphs have been deleted and rewritten due to requests from other reviewers. 
